# OpenReview forum: "Playing Language Game with LLMs Leads to Jailbreaking"
_ICLR.cc/2025/Conference — ICLR 2025 Conference Withdrawn Submission_

### Official Review · Reviewer_cdZ9 · 2024-10-27

**Soundness:** 2
**Presentation:** 3
**Contribution:** 2
**Rating:** 3
**Confidence:** 4

**Summary:**

This paper proposes novel LLM jailbreaking methods via *playing language games with LLMs*. The authors both consider natural language games (that already exist) and design novel custom language games. The results show that SOTA proprietary LLMs are vulnerable against such jailbreaking attempts.

**Strengths:**

* The proposed attack strategies are novel, utilizing simple language games.
* More interestingly, the authors also study whether safety fine-tuning against one language game attack could help the model resist other custom language game attacks.

**Weaknesses:**

* Need more implementation details on experimental setup. For example, what's the temperature of the three LLMs you evaluated against? Also, I would like to see repetitive experiment results (or error bars).

  In Line 360-367, you mentioned that "GPT-4o-mini exhibited different success rates for Self 4 (**86\%**) and Self 5 (**82\%**)." This seems like a small variations. Are you sure this is not caused by the randomness during language model decoding?

* Need to validate the claims on more models. For now, the experiments only demonstrate effectiveness on 3 proprietary models. Would be necessary to report results against more models (e.g., Llama-3-405B or other smaller open-weight models). More results are good to know, even if these models are not capable enough to play the language games.

* Need more details on how you conducted the safety alignment fine-tuning (Sec 4.5). Currently it's quite vague. For example, how is the jailbreak dataset constructed?

* What if you fine-tune the Llama-3.1-70B model over more custom language games (say Self 1- Self 4)? I wonder whether safety training over multiple language games can help the safety refusal behaviors generalize better on unseen language game attacks.

  While I appreciate the authors' efforts on exploring the potential defense by conducting safety fine-tuning over a single language game attack, it would be interesting to see whether fine-tuning over more language game attacks could allow better generalizations and thus help real-world model developers defend against the proposed attacks.

* May need to evaluate the proposed attacks against some existing defense strategies [1-3].

* How much would the model utility drop if uers chat with LLMs under these language games? I think the authors can evaluate the models on some utility benchmarks (e.g., MT-Bench) when playing the language games, in order to show that the models can still provide useful responses in general.

* I feel the contribution of this work is somehow diminished when compare it to the existing work [4] that jailbreak LLMs via encryption and encoding (though the authors indeed discussed the difference).

[1] Baseline defenses for adversarial attacks against aligned language models, Arxiv 2023

[2] Defending chatgpt against jailbreak attack via self-reminders, NMI 2023

[3] Smoothllm: Defending large language models against jailbreaking attacks, Arxiv 2023

[4] Gpt-4 is too smart to be safe: Stealthy chat with llms via cipher, ICLR 2024

**Questions:**

* I still wonder why "human redability" is important for a jailbreak attack? May need better justification for this point.
* Line 435-436: "Notably, the fine-tuned model was able to successfully defend against **other forms of attacks**." Is this a typo? Do you mean that the fine-tuned model can defend well against the attack which was considered in the fine-tuning dataset (but not the others)?
* Why do you use GPT-4o-mini as the judge, but not MD-Judge which comes along with SALAD-Bench? Also, why translate to Chinese during jailbreak judgments?
* I'm not following well from Line 322 to Line 327. What do you mean by "GPT-4o and GPT-4o-mini often address questions while framing their answers in a positive manner?" Why is this bad? Can you show some qualitative examples to justify this?

---

### Official Review · Reviewer_MKGg · 2024-11-01

**Soundness:** 2
**Presentation:** 2
**Contribution:** 2
**Rating:** 3
**Confidence:** 4

**Summary:**

The paper studies the usage of "language games" in jailbreaking large closed-source LLMs. Language games are essentially conversational games played with a LLM that follow some simple rules (e.g. character replacement, insertion, etc.) The paper demonstrates that through the usage of language games, a user may jailbreak existing models. The paper also finds that larger models are *more* susceptible to the attack than smaller ones.
Finally, the paper fine-tunes a Llama model to defend against the attack, and alarmingly demonstrates that fine-tuning against a single language game does not grant defense against other language games.

**Strengths:**

The proposed method is simple, and easy to follow. The method itself demonstrates a new class of easy to create jailbreaks which may be difficult to defend against. Moreover, the result that larger models are more susceptible to the attack is interesting, and presents a potential challenge for alignment of future models.

**Weaknesses:**

- I am extremely concerned about the novelty and evaluation of the attack. Prior work [1, 2, 4, 5] suggests different techniques to modify the prompt to jailbreak the model. While the paper cites several of these methods, it does not compare against them; it is unclear if the proposed attack is more concerning than those already presented in prior work. In particular, I would like to know if the claim made by the paper (that larger models are *more* susceptible to this class of attack) is true for other attacks as well (this could be a very interesting result!).

- The evaluation also lacks comparison to a baseline (i.e. where no language game was used). It is unclear how susceptible the base models themselves are to the subset of prompts used during evaluation.

- I am concerned about the usage of gpt-4o mini as an evaluator; Cross-validation of the judge compared to prior methods (e.g. MD-Judge from SALAD-Bench) would be helpful in establishing its correctness. Additionally, using closed-source models as evaluators may not be reproducible, as their behavior is black-box and may change over time [3].

- The results showing lack of generalization of the defense in fine-tuned Llama 70b are interesting! The paper claims that larger models are more susceptible, so it would be interesting to compare to a fine-tuned variant of a smaller model (e.g. Llama 3.1 8b).

[1] Bianchi, Federico, Mirac Suzgun, Giuseppe Attanasio, Paul Röttger, Dan Jurafsky, Tatsunori Hashimoto, and James Zou. “SAFETY-TUNED LLAMAS: LESSONS FROM IMPROV- ING THE SAFETY OF LARGE LANGUAGE MODELS THAT FOLLOW INSTRUCTIONS,” 2024.

[2] Deng, Yue, Wenxuan Zhang, Sinno Jialin Pan, and Lidong Bing. “Multilingual Jailbreak Challenges in Large Language Models.” arXiv, March 4, 2024. http://arxiv.org/abs/2310.06474.

[3] Xie, Tinghao, Xiangyu Qi, Yi Zeng, Yangsibo Huang, Udari Madhushani Sehwag, Kaixuan Huang, Luxi He, et al. “SORRY-Bench: Systematically Evaluating Large Language Model Safety Refusal Behaviors.” arXiv, June 20, 2024. http://arxiv.org/abs/2406.14598.

[4] Zeng, Yi, Hongpeng Lin, Jingwen Zhang, Diyi Yang, Ruoxi Jia, and Weiyan Shi. “How Johnny Can Persuade LLMs to Jailbreak Them: Rethinking Persuasion to Challenge AI Safety by Humanizing LLMs.” arXiv, January 23, 2024. https://doi.org/10.48550/arXiv.2401.06373.

[5] Zou, Andy, Zifan Wang, Nicholas Carlini, Milad Nasr, J. Zico Kolter, and Matt Fredrikson. “Universal and Transferable Adversarial Attacks on Aligned Language Models.” arXiv, December 20, 2023. https://doi.org/10.48550/arXiv.2307.15043.

**Questions:**

- I noticed the evaluation prompt used by gpt-4o mini first translates to Chinese (A.1) . Why is this needed? And does this impact the judge at all?

- Have you compared to a simple random baseline? (e.g. randomly modifying or inserting characters?)

- What happens to success rates when the language game prompt is not present?

- When fine-tuning the model, was the language game prompt included?

---

### Official Review · Reviewer_G43c · 2024-11-02

**Soundness:** 1
**Presentation:** 3
**Contribution:** 3
**Rating:** 3
**Confidence:** 4

**Summary:**

This paper explores how language games can be used to jailbreak LLMs, where each game is comprised of a set of rules for how to transform a text prompt. The authors argue that these language games give rise to a class of jailbreaks that 1. Even when the LLM makes errors in applying the rules of the game, it is still easy for a human to infer how these errors are to be corrected due to the natural language nature of the game, and 2. Are easy to construct, such that if the LLM is fine-tuned to be robust against one language game, it’s relatively easy to create a new language game jailbreak that still bypasses safety. Both previously known and novel language games are evaluated on GPT-4 and Claude 3.5 models, and the authors report that the proposed jailbreak technique is successful against these models. The authors also investigate safety generalization across language games when the model is fine-tuned to be robust against a specific language game, and find that generalization in general is quite poor, even between very similar games. This is in agreement with the conjecture that successful jailbreaks against aligned LLMs may be explained by the phenomenon of mismatched generalization [1].

[1] Wei, A., Haghtalab, N. and Steinhardt, J., 2024. Jailbroken: How does llm safety training fail? Advances in Neural Information Processing Systems, 36.

**Strengths:**

1. The proposed jailbreaking technique is easy to understand and easy to create variations of (i.e., the custom language games.)
2. It is good that the proposed attacks are tested on SoTA closed-source models; this indeed shows that these attacks have practical, real-world safety consequences.
3. The results in table 5 are interesting; they demonstrate that safety generalization after fine-tuning to be robust against specific games is poor even to extremely similar games. To the best of my knowledge previous claims of poor safety generalization to other jailbreaks usually just consider jailbreaks that are very different [1], so this is an interesting finding.

[1] Mazeika, Mantas, et al. "Harmbench: A standardized evaluation framework for automated red teaming and robust refusal." arXiv preprint arXiv:2402.04249 (2024).

**Weaknesses:**

1. Line 284-285: “Each model is in its default settings to ensure consistency and to simulate real-world use cases.” One downside of this is that sampling is used during the decoding process (as opposed to using greedy decoding), meaning the results are not quite reproducible. I would suggest the authors either 1. Report results using greedy decoding and a fixed seed, or 2. If using sampling, report some summary statistic (e.g., average with standard deviation) over multiple runs (ideally, also with a fixed seed). This may also help clarify whether the table 3 observation that similar language games can have different behaviors is more so due to randomness or if there is a genuine difference in behavior.
2. There is no measurement of how reliable the GPT-4o-mini-based safety judge is. It may strengthen the results to add some human evaluation to measure reliability, e.g., by reporting false positive/negative rates on a sample of data points.
3.  The method of translation seems a bit unreliable. For example, in figure 8, "Certalfainly! Halerfe alfare stalfealfe” is translated to “rtainly! halerfe ar stealf,” but if the rules of the game are followed it should’ve been translated to “Certainly! Here are stee.” Doesn’t it make more sense to just programatically perform the translation (e.g., for alfa balfa, just remove instances of “alf” after the first consonant/before vowels) before being evaluated by the safety judge? Otherwise, this just adds another source of error that may drastically affect your reported evaluation results.

I would be willing to raise my score if these points are addressed and the results are found to be convincing.

**Questions:**

1. Lines 160-161: “However, while these methods … is generalized across the intermediate layers of LLMs.” This seems abrupt — no discussion about the role of intermediate layers had been introduced before this. Is this point more about mismatched generalization of the input data?
2. Can you compare/contrast natural language games with ciphers (e.g., with those found in Yuan et al., 2023)? It seems they are similar in spirit as ciphers also apply various rules for transforming the input, so the differences seem less clear to me. It may be helpful to readers to make this clear in the paper.
3. Line 214: “The models used in the experiment possess prior knowledge of these types of linguistic manipulations.” Can you provide some more information about this? I agree this is probably the case, but I’m just curious whether there are sources backing this up or if it remains just a conjecture (if so, it should probably be rephrased as such), given these models are closed-source and that their training data details have not been released.
4. Please clarify why the prompt in Appendix A.1 has the model perform translation to and from Chinese; it is never explained in the paper. Also, the prompt only asks to provide a 1 or a 0 as the judgement and contains no mention about “unclear” labeling, so how are unclear results determined? Lastly, the referenced figure on line 598 is incorrect (16 instead of 4).
5. Lines 323-327: can you provide specific case studies for these claims? For example, the case studies in the appendix don’t seem to have examples of unclear responses for GPT-4o/4o-mini.
6. For each case study in the appendix, can you also provide the labels that were given by the safety judge?
7. Line 435-436: “Notably, the fine-tuned model was able to successfully defend against other forms of attacks, with a success rate of 0% to 3%.” Do you mean to say defending against the attack it was fine-tuned on?
8. (Optional, just curious) It could be interesting to see if safety generalization to other language games can be achieved by fine-tuning against multiple language games at a time. Is it possible that at some point (i.e., with a sufficient amount of games in the fine-tuning set), the model is able to overcome shortcut learning/overfitting to specific games?

---

### Official Review · Reviewer_UM8Y · 2024-11-03

**Soundness:** 1
**Presentation:** 1
**Contribution:** 1
**Rating:** 1
**Confidence:** 4

**Summary:**

This paper presents a jailbreak technique leveraging language games to obfuscate malicious prompts through, e.g., inserting "ub" or "ob" before syllable rimes. The authors propose both natural and custom language game variants, evaluating their approach on three language models using a single dataset, achieving jailbreak success rates up to 93%.

**Strengths:**

The proposed approach demonstrates that simple linguistic transformations can effectively circumvent LLM safety measures, providing valuable insights into model vulnerabilities.

**Weaknesses:**

1. The approach lacks systematic methodology, relying heavily on manual crafting of language games. A more principled framework for automatically generating and adapting language patterns would strengthen the contribution.

2. The empirical evaluation requires significant expansion. The authors should include additional benchmarks beyond a single dataset (e.g., AdvBench[1]) to demonstrate generalizability, and evaluate on more models like Llama-2 to show broader applicability.

3. While the fine-tuning experiments are valuable, baseline results on vanilla Llama3 would provide important context for understanding the effectiveness of this defense approach.

4. The paper would benefit from comprehensive comparisons with current state-of-the-art attacks such as GCG[1], AutoDAN[2], PAIR[3], TAP[4], DeepInception[5], as well as how robust the proposed method is against defenses such as  paraphrasing[6], SmoothLLM [7], Backtranslation[8].

[1] Andy Zou, Zifan Wang, J. Zico Kolter, and Matt Fredrikson. Universal and transferable adversarial
attacks on aligned language models. CoRR, abs/2307.15043, 2023. doi: 10.48550/ARXIV.2307.
15043. URL https://doi.org/10.48550/arXiv.2307.15043.

[2] Xiaogeng Liu, Nan Xu, Muhao Chen, and Chaowei Xiao. AutoDAN: Generating stealthy jailbreak
prompts on aligned large language models. In The Twelfth International Conference on Learning
Representations, 2024. URL https://openreview.net/forum?id=7Jwpw4qKkb

[3] Patrick Chao, Alexander Robey, Edgar Dobriban, Hamed Hassani, George J. Pappas, and Eric Wong.
Jailbreaking black box large language models in twenty queries. CoRR, abs/2310.08419, 2023.
doi: 10.48550/ARXIV.2310.08419. URL https://doi.org/10.48550/arXiv.2310.
08419.

[4] Anay Mehrotra, Manolis Zampetakis, Paul Kassianik, Blaine Nelson, Hyrum Anderson, Yaron
Singer, and Amin Karbasi. Tree of attacks: Jailbreaking black-box llms automatically. CoRR,
abs/2312.02119, 2023. doi: 10.48550/ARXIV.2312.02119. URL https://doi.org/10.
48550/arXiv.2312.02119

[5] Xuan Li, Zhanke Zhou, Jianing Zhu, Jiangchao Yao, Tongliang Liu, and Bo Han. Deepinception:
Hypnotize large language model to be jailbreaker. CoRR, abs/2311.03191, 2023. doi: 10.48550/
ARXIV.2311.03191. URL https://doi.org/10.48550/arXiv.2311.03191.

[6] Jain, Neel, et al. "Baseline defenses for adversarial attacks against aligned language models." arXiv preprint arXiv:2309.00614 (2023).

[7] Robey, Alexander, et al. "Smoothllm: Defending large language models against jailbreaking attacks." arXiv preprint arXiv:2310.03684 (2023).

[8] Wang, Yihan, et al. "Defending llms against jailbreaking attacks via backtranslation." arXiv preprint arXiv:2402.16459 (2024).

**Questions:**

The paper contains ambiguous terminology that needs clarification. For example, in discussing defense results, the distinction between "other forms of attacks" (0-3% success) and "other custom language games" (failed defense) is unclear. Please specify which attack categories these refer to.

---

### Note · Authors · 2024-11-16

I have read and agree with the venue's withdrawal policy on behalf of myself and my co-authors.